# FREE-MAD: CONSENSUS-FREE MULTI-AGENT DEBATE

## ABSTRACT

Multi-agent debate (MAD) is an emerging approach to improving the reasoning capabilities of large language models (LLMs). Existing MAD methods rely on multiple rounds of interaction among agents to reach consensus, and the final output is decided by majority voting in the last round. However, this consensus-based design faces several limitations. First, multiple rounds of communication increases token overhead and limits scalability. Second, due to the inherent conformity of LLMs, agents that initially produce correct responses may be influenced by incorrect ones during the debate process, causing error propagation. Third, majority voting introduces randomness and unfairness in the decision-making phase, and can degrade the reasoning performance. To address these issues, we propose FREE-MAD, an alternative and novel MAD framework that eliminates the need for consensus among agents. FREE-MAD introduces a novel score-based decision mechanism that evaluates the entire debate trajectory rather than relying on the last round only. This mechanism tracks how each agent's reasoning evolves, enabling more accurate and fair outcomes. In addition, FREE-MAD reconstructs the debate phase by introducing anti-conformity, a mechanism that enables agents to mitigate excessive influence from the majority. Experiments on eight benchmark datasets demonstrate that FREE-MAD significantly improves reasoning performance while requiring only a single-round debate and thus reducing token costs. We also show that compared to existing MAD approaches, FREE-MAD exhibits improved robustness in real-world attack scenarios.

## 1 INTRODUCTION

Large language models (LLMs), due to their strong reasoning capabilities, have been widely applied in domains such as chatbots (Li et al., 2024a), programming (Yan et al., 2024), healthcare (Liu et al., 2024a), and cybersecurity (Zou et al., 2024). Recent applications of LLM agents (Luo et al., 2025b) have placed increasing demands on their reasoning performance. To improve the reasoning accuracy of LLM agents, many studies have explored training-free methods such as Chain-of-Thought (CoT) (Wei et al., 2022), self-refinement (Madaan et al., 2023) and self-consistency (Wang et al., 2023). However, these methods focus on the usage of single LLMs and lack collaboration among multiple models. In contrast, multi-agent debate (MAD) (Zeng et al., 2025; Du et al., 2024; Chan et al., 2024; Liu et al., 2025) has emerged as a prominent solution, showing that multiple agents engaged in multi-round interactions can achieve substantially better reasoning performance than a single agent. Indeed, MAD has been used in various scenarios such as software issue resolution (Li et al., 2025), mathematical reasoning (Zhang & Xiong, 2025), and code summarization (Chun et al., 2025).

Existing efforts to optimize MAD focus primarily on the reasoning strategies of individual agents during debates (Liu et al., 2025) or improving scalability (Zeng et al., 2025). Meanwhile, recent studies reveal that LLM agents can exhibit conformity (Weng et al., 2025; Zhu et al., 2025; Cho et al., 2025), meaning that during multi-agent interactions, agents tend to favor answers endorsed by the majority. In existing MAD frameworks, conformity-driven consensus (Sun et al., 2024; Zeng et al., 2025; Li et al., 2024b) is employed during the debate process to obtain the correct answer as the final decision (Chan et al., 2024). However, such consensus reduces reasoning accuracy. The consensus-based MAD schemes suffer from the Silent Agreement problem (Wang et al., 2025b). Even when the agents start with divergent opinions, they remain silent during the discussion due to conformity.

As a result, the agent group eventually provides an incorrect answer. More importantly, consensus naturally demands more debate rounds, increasing token consumption and limiting scalability.

To address these limitations, we propose FREE-MAD, a consensus-free MAD framework that reconstructs both the debate stage and the decision stage. In the debate stage, we integrate the conventional conformity mode as used in prior work with a new mode called anti-conformity. In particular, the anti-conformity mode leverages CoT to encourage agents to identify flaws in the outputs from other agents; this situation is in contrast to existing approaches that use consensus as an indicator of correctness. In the decision stage, we propose a score-based mechanism that evaluates *all* intermediate outputs across debate rounds, instead of focusing solely on the final round as in traditional MAD frameworks. By tracking changes in the reasoning trajectories of all agents, FREE-MAD assigns scores to all candidate responses without requiring consensus in the debate stage. Furthermore, we theoretically formalize the MAD protocol, enabling a formal comparison between FREE-MAD and existing MAD variants.

To evaluate FREE-MAD, we have conducted extensive experiments on eight benchmark datasets, covering knowledge-based reasoning, logical reasoning, and mathematical reasoning with varying levels of difficulty. We show that FREE-MAD outpaces baseline approaches in terms of reasoning accuracy. In particular, FREE-MAD achieves improved accuracy with fewer debate rounds, thus accelerating the debate process while maintaining strong scalability. Furthermore, we show that FREE-MAD exhibits enhanced robustness (Chen et al., 2024a) and maintains accurate reasoning against communication attacks (He et al., 2025), where interactions among agents might be partially disrupted. We summarize our contributions as follows:

- We propose a novel consensus-free MAD framework, called FREE-MAD, with dedicated optimizations in both the debate and decision stages. We design a score-based decision mechanism that evaluates all intermediate results across debate rounds, enabling accurate reasoning without requiring consensus.
- We provide a syntax for the MAD protocol and perform a formal analysis of FREE-MAD and existing mechanisms.
- We implement FREE-MAD and conduct extensive experiments on eight benchmarks. We show that FREE-MAD outperforms existing protocols in terms of reasoning accuracy, scalability, and robustness.

## 2 RELATED WORK

**MAD Protocols**. When MAD was introduced, it was intended to improve reasoning by encouraging LLMs to reach consensus across their answers. However, such a consensus cannot be guaranteed in theory and is generally achievable only in practice (Du et al., 2024). Moreover, consensus becomes easier to achieve when agents adjust their trust between self-generated and externally generated content, yet this adjustment tends to reduce reasoning accuracy. The researchers did not investigate the deeper conformity issue that underlies this performance drop. Subsequent studies have focused on optimizing the debate stage of MAD (Chan et al., 2024; Chen et al., 2024b; Zeng et al., 2025; Liang et al., 2024; Liu et al., 2025). In these works, consensus is commonly treated as the default goal of the debate stage (Li et al., 2024b). In this paper, we show that MAD can operate effectively without requiring consensus.

**MAD Protocol Security**. In the debate stage, traditional consensus-oriented approaches for agents tend to lack robustness in unreliable network environments. Communication attacks (He et al., 2025) can force some agents to withdraw from the debate, preventing them from receiving others' responses. This delays consensus and increases overhead in adaptive schemes (Liang et al., 2024), ultimately reducing reasoning accuracy. Another line of work allows agents to return both their individual responses and a self-assessed confidence score (Chen et al., 2024b), which is then used in a weighted aggregation of the final result. However, due to LLM hallucinations (Ji et al., 2023), such confidence may be unreliable. Existing decision mechanisms in MAD are inadequate for addressing the security risks associated with LLM conformity. In real-world deployments, if a small subset of agents is compromised via prompt injection attacks (Greshake et al., 2023; Liu et al., 2024b; Zhan et al., 2025), the system may converge toward a shared but incorrect answer. When decisions are made using mechanisms such as majority voting, this can lead to a complete failure of the MAD system. Other

approaches (Liang et al., 2024) use an LLM-as-a-Judge (Zheng et al., 2023) framework, where an LLM decides the final outcome. This approach can produce biased results and is prone to conformity, making it effectively equivalent to majority voting. In addition, if the LLM is compromised by a prompt injection attack, the accuracy of the MAD system's final output may drop significantly. We discuss related work on agent diversity in MAD further in Appendix B.

## 3 PRELIMINARY ANALYSIS

### 3.1 MAD PROTOCOL: A FORMAL TWO-PHASE DECOMPOSITION

To enable a formal analysis, we decompose the MAD protocol into two core stages: **Debate** and **Decision**. The Debate stage internally unfolds over $R$ iterative rounds, culminating in a set of final answers. Formally, given a set of $N$ agents denoted as $\{a_i\}_{i=1}^N$, the protocol is defined as:

$$\{r_i^R\}_{i=1}^N \leftarrow \texttt{Debate}(\{a_i\}_{i=1}^N, q, p, R), \tag{1}$$

$$r_{\text{final}} \leftarrow \texttt{Decide}(\{r_i^R\}_{i=1}^N). \tag{2}$$

In the **Debate** stage, all agents engage in a multi-round interaction based on a user query $q$ and a guiding prompt $p$ that specifies how agents should debate. The debate unfolds over $R$ rounds. The debate begins with an initial step where each agent generates a preliminary response to $q$, which is then broadcast to all other agents as an auxiliary context (Yang et al., 2025). Subsequently, each agent $a_i$ iteratively updates its own answer $r_i^k$ over $R$ rounds, resulting in a final set of responses $\{r_i^R\}_{i=1}^N$. The complete history of utterances up to round $k-1$ is denoted as the context $C^{(k-1)}$. We model LLM agents as a probabilistic process to capture their generative behavior. In round $k$, agent $a_i$ produces its response $r_i^k$ by sampling from a conditional probability distribution defined over the preceding context and $p$:

$$r_i^k \sim P_{a_i}(r|C^{(k-1)}, p). \tag{3}$$

To study the interplay between independent reasoning and conformity, we model the overall probability distribution $P_{a_i}$ with a formulation that separates the contributions of the two factors:

$$P_{a_i}(r|C^{(k-1)}, p) = \frac{1}{Z} \cdot P_{\text{in}}(r|q, p) \cdot \exp(\beta(p) \cdot S_{\text{con}}(r, C^{(k-1)})), \tag{4}$$

where the independent reasoning distribution $P_{\text{in}}(r \mid q, p)$ mathematically characterizes the agent's intrinsic reasoning ability given the question $q$ and prompt $p$. This ability excludes peer influence. The conformity score $S_{\text{con}}(r, C^{(k-1)})$ measures how much a candidate response $r$ aligns with peer utterances in $C^{(k-1)}$. Its effect is scaled by the conformity parameter $\beta(p)$, which is determined by $p$. By default, LLMs exhibit a tendency to conform. We capture this by assuming $\beta_{\text{default}} > 0$ when $p$ is empty or neutral. If $p$ encourages critical reasoning, $\beta(p)$ can be negative, acting as a regularizer that reduces alignment and pushes responses toward $P_{\text{in}}$. Conversely, if $p$ encourages agreement with majority opinions, $\beta(p)$ assumes a larger positive value, which speeds up convergence towards consensus. Finally, the model combines all components via an exponential transformation and normalizes by $Z$ to ensure a valid probability distribution.

In the **Decision** stage, a final output $r_{\text{final}}$ is selected from $\{r_i^R\}_{i=1}^N$, typically through mechanisms such as majority voting. In this work, $R$ excludes initial response generation and begins once agents start receiving responses from others.

### 3.2 WEAKNESSES OF EXISTING MAD APPROACHES

**Reasoning Accuracy**. MAD approaches (Du et al., 2024; Yang et al., 2025; Li et al., 2024b) design the decision stage[1] to operate on the final round's $N$ responses in the debate, while overlooking the remaining $R \times N$ intermediate responses that emerged throughout the debate process. This omission diminishes the influence of these earlier responses on determining $r_{\text{final}}$, thereby reducing

---

[1] In this work, we focus exclusively on scenarios where only one option can be selected.

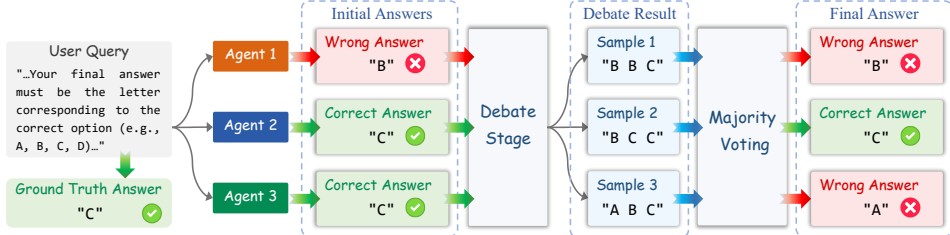

Figure 1: Existing MAD approaches may obtain final answers that are even less accurate than the initial ones.

both the accuracy and fairness of the final outcome. In consensus-based debates, the process ends once the agents reach agreement, even if the answer is incorrect (Chen et al., 2024b; Wang et al., 2025b). However, Du et al. (2024) has shown that it is still possible for the correct answer to emerge during later stages of the debate, even if none of the agents initially generates a correct answer. Early termination thus reduces MAD's problem-solving accuracy. From empirical observations, we find that the initial responses generated independently by multiple agents may outperform the debate results obtained after applying MAD. As shown in Figure 1, applying majority voting directly to initial answers can yield the correct result, while debate outcomes may be incorrect. We illustrate three possibilities. In Sample 1, the agents reach consensus on an incorrect answer. Sample 2 represents the desired outcome, where the correct answer "C" holds the majority. Outcomes such as Sample 3 have received little attention. The set $\{r_i\}_{i=1}^N$ contains entirely distinct outputs, with no repetitions or equal counts for multiple answers. Under these circumstances, the final answer is determined either by selecting a response at random from the set $\{r_i\}_{i=1}^N$ or by choosing the first one. Both strategies substantially degrade the accuracy of MAD. Therefore, majority voting is unsuitable for decisions based on debate outcomes. More robust and practical mechanisms are required.

**Robustness**. Most existing multi-agent systems lack robustness (Chen et al., 2024a; Zhang et al., 2024; He et al., 2025). MAD is even more vulnerable to attacks (Qi et al., 2025) due to its consensus mechanism. Researchers (Luo et al., 2025a) mainly enhance the robustness of multi-agent systems by introducing blockchain and leveraging its traditional distributed consensus protocols. However, the use of blockchain, including smart contracts (Li et al., 2023) and consensus execution (Zhang et al., 2023; Duan et al., 2018), drastically impacts system performance.

## 4 METHODOLOGY

In this section, we first provide a formalization to introduce our proposed FREE-MAD (see Section 4.1). We then present the two core techniques within FREE-MAD: a debate protocol that operates without requiring consensus (see Section 4.2), and a score-based decision mechanism (see Section 4.3). Then, we provide a description of FREE-MAD (see Section 4.4). Finally, we conduct an analysis between FREE-MAD and existing approaches (see Section 4.5).

### 4.1 FREE-MAD

FREE-MAD focuses on the complete set of outcomes generated throughout the entire MAD process, rather than limiting attention to only the final-round responses, as done in traditional approaches. Our proposed debate protocol incorporates all responses into the decision-making process. This perspective can be formally represented using the following matrix formulation:

$$\texttt{Decide}[\alpha_0, \alpha_1, \ldots, \alpha_{R-1}, \alpha_R] = \begin{bmatrix} r_1^0 & r_1^1 & \cdots & r_1^{R-1} & r_1^R \\ \vdots & \vdots & \ddots & \vdots & \vdots \\ r_N^0 & r_N^1 & \cdots & r_N^{R-1} & r_N^R \end{bmatrix} \to r_{\text{final}}. \tag{5}$$

The highlighted entries $(\alpha_0, \alpha_1, \ldots, \alpha_{R-1})$ represent components that were not considered in previous MAD approaches. $r_i^0$ denotes the initial response generated by agent $i$. Unlike prior MAD frameworks that decouple debate and decision stages, FREE-MAD innovatively integrates decision-making into the debate phase, resulting in a unified process (see Figure 2).

## 4.2 CONSENSUS-FREE DEBATE

The agents in MAD exchange responses and update their answers based on a prefix prompt. A common prompt is "The responses from other agents are as follows" (Du et al., 2024). Due to the conformity of LLMs, such prompts often bias the agent toward the majority answer. This undermines rigorous reasoning and raises the likelihood that reasoning errors remain unnoticed. To mitigate this conformity, we design an additional CoT-based prompt (see Appendix E) appended after the prefix prompt, motivated by (Weng et al., 2025). We provide agents with a scenario that includes adversarial agents. We instruct agents to carefully assess the discrepancies between their own answers and those from peers. Agents are expected to change their beliefs only if there is a clear indication that their own answer is incorrect, rather than aiming to reach consensus with others. This mechanism is intended to reduce the propagation of incorrect answers during the MAD process.

Our structured and critical reasoning prompt forms the core mechanism for optimizing the probabilistic model (Equation 4) and has two main effects. First, it improves the quality of the independent reasoning distribution $P_{\text{in}}$. FREE-MAD requires each agent to provide an answer along with a detailed reasoning trace, which is incorporated into the next-round context $C^{(k-1)}$. Agent $a_i$ uses this context to analyze peers' reasoning rather than just their answers. If the reasoning behind the majority answer is flawed, $a_i$'s own critical thinking will assign a very low probability to that answer in $P_{\text{in}}$. Second, it balances conformity. Even if a popular but incorrect answer has a high $S_{\text{con}}$, a low $P_{\text{in}}$ keeps its overall probability $P_{a_i}$ low.

---

**Algorithm 1:** MAD Protocol via Score-Based Decision and its Evaluation

---

**Input:** Answer matrix $A \in \mathbb{R}^{N \times (R+1)}$ from $N$ agents over $R$ rounds; List of task inputs and ground truth responses $\mathcal{D}_{\text{task}} = \{q_i, r_i^g\}$; Weights $\mathcal{W} = \{w_i\}_{i=1}^4$; Guiding prompt $p$

**Output:** Accuracy

success $\leftarrow$ 0; Initialize empty score dictionary $S \leftarrow \emptyset$

**for** $q, r^g$ in $\mathcal{D}_{task}$ **do**

  **for** $k \leftarrow 0$ **to** $R$ **do**

    $f = (k+1)^{-1}$ # Initial scoring factor with a non-zero value.

    **for** $i \leftarrow 1$ **to** $N$ **do**

      **if** $k \neq 0$ **then**

        Context $C \leftarrow$ Aggregate responses from other agents in round $k-1$

      $r_i^k \leftarrow P_{a_i}(q, p)$; Update $C$ and $A$ with $r_i^k$

      $\hat{r} \leftarrow A[i][k]$

      **if** $k = 0$ **then**

        $S[\hat{r}] \leftarrow S[\hat{r}] + w_1 f$ # Assign an initial score to the answer.

      **else**

        $r_p \leftarrow A[i][k-1]$ # Find the answer of agent $a_i$ in the previous round.

        **if** $\hat{r} \neq r_p$ **then**

          **if** $r_p \in S$ **then**

            $S[r_p] \leftarrow S[r_p] - w_2 f$ # The answer has been transferred from $r_p$ to $\hat{r}$.

          $S[\hat{r}] \leftarrow S[\hat{r}] + w_3 f$

        **else**

          $S[\hat{r}] \leftarrow S[\hat{r}] + w_4 f$ # The agent $a_i$ maintains this answer.

  Remove keys from $S$ where key is None.

  $MS \leftarrow \arg\max_k S[k]$

  **if** $|MS| > 1$ **then**

    Randomly choose $r_{\text{final}} \in MS$ # Guarantee robustness.

  **else**

    $r_{\text{final}} \leftarrow MS[0]$

  success $\leftarrow$ success $+ \mathbb{I}(r_{\text{final}} = r^g)$

**Return** $\frac{\text{success}}{|\mathcal{D}|}$

---

## 4.3 SCORE-BASED DECISION MECHANISM

Based on the existing MAD framework (Du et al., 2024) and formalization of the traditional MAD protocol (Subramaniam et al., 2025), we describe our protocol as shown in Algorithm 1. This method

maintains a matrix $A \in \mathbb{R}^{N \times (R+1)}$ that records the real-time responses of each agent across debate rounds. Concurrently, a score dictionary $S$ is used to track the scores associated with multiple answers that emerge throughout the debate. The mechanism evaluates the likelihood of an answer being correct by tracking whether agents exhibit a shift in their opinions across rounds (whether the answer provided in the current round differs from that in the previous round). The agents in this framework are not designed to seek consensus; instead, they rigorously assess the reasoning behind the answers. Therefore, a change in an agent's response is interpreted as an indicator that a more accurate answer has been identified. Specifically, answers that agents abandon are considered more likely to be incorrect, whereas newly adopted answers are treated as more likely to be correct. This dynamic informs the score updates within the dictionary $S$.

We assign different weights $w_i f$ to answers based on whether agents have changed their responses between rounds. Here, $w_i \in \mathcal{W}$ represents system-defined parameters, and $f$ is a correction factor inversely proportional to the current round number. As the number of rounds increases, the amount of contextual information each agent receives also grows, thereby increasing the risk of confor-

Table 1: A comprehensive comparison between our proposed approach and existing methods.

| Approaches | Decision | Anti-Conformity | Fairness | Security |
|---|---|---|---|---|
| Sparse MADLi et al. (2024b) | Majority Voting | ✗ | ✗ | ✗ |
| Liang et al. (2024) | LLM-as-a-Judge | ✗ | ✗ | ✗ |
| ReConcile (Chen et al., 2024b) | Weighted Voting | ✗ | ✓ | ✗ |
| ChatEval (Chan et al., 2024) | Majority Voting | ✗ | ✗ | ✗ |
| DMAD (Liu et al., 2025) | Majority Voting | ✗ | ✗ | ✓ |
| SoM (Du et al., 2024) | Majority Voting | ✗ | ✓ | ✓ |
| FREE-MAD-N | score | ✓ | ✓ | ✓ |
| FREE-MAD-C | score | ✗ | ✓ | ✓ |

mity, particularly for agents based on smaller models. To limit the influence of conformity, the impact of opinion shifts in later rounds is downweighted via the factor $f$. We define $MS$ as the set of answers with the highest score. Although this set usually contains a single candidate, we adopt a randomized selection strategy to maintain theoretical robustness.

## 4.4 FRAMEWORK DESIGN

As noted in (Weng et al., 2025), LLMs' conformity presents a double-edged sword. On the one hand, it fosters consensus and cohesive outcomes. On the other hand, it undermines the reliability of agents' judgments in sensitive domains such as voting. Debate based on anti-conformity mitigates the negative effects of blind conformity among agents. However, for relatively simple tasks, LLMs may show excessive anti-conformity, which causes stubbornness and reduces reasoning accuracy. Therefore, we argue that conformity-oriented and anti-conformity-oriented debate modes should be adapted and switched according to the task. To achieve finer control over excessive conformity, the weight parameter $\mathcal{W}$ in the score mechanism can be adjusted, which regulates the balance between an agent changing its opinion and maintaining its original stance. Based on this idea, we pro-

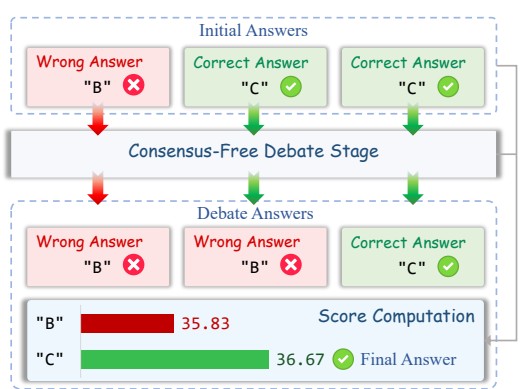

Figure 2: The inference process of our MAD framework. When the correct answers are in the minority in the final round, the framework is still able to identify the correct one as the final answer.

pose FREE-MAD, which consists of two sub-schemes that share a score-based decision stage. FREE-MAD-N integrates an anti-conformity debate process, while FREE-MAD-C integrates a conformity-based debate process. Together, the schemes extend the framework's applicability to a wider range of real-world scenarios.

## 4.5 ANALYSIS

We compare our proposed FREE-MAD framework with existing MAD approaches, as shown in Table 1. Below, we provide a detailed analysis of the advantages of FREE-MAD.

**Accuracy**. The FREE-MAD framework adopts a consensus-free debate strategy, which helps reduce the influence of conformity. This approach encourages agents to concentrate more on the correctness of reasoning steps, and may alleviate the effect of error propagation. In addition, existing methods usually require multiple rounds of debate to reach consensus. When the number of rounds $R$ is small, such as when $R = 1$, these approaches may experience reduced reasoning performance due to the lack of consensus, which can lead to lower accuracy. In contrast, the performance of FREE-MAD is not closely tied to $R$.

**Scalability**. Prior work (Zeng et al., 2025) provides a general complexity analysis of the token cost (TC) in MAD as: $\mathcal{O}\big(NR^2V + N^2RV + NR|q|\big)$, where $V$ is the maximum token cost for each agent. Therefore, a practical MAD framework should aim to achieve high accuracy with fewer agents and fewer rounds (Liu et al., 2025). The consensus-building process generally takes 2 to 3 rounds (Chen et al., 2024b; Du et al., 2024; Yang et al., 2025; Xiong et al., 2023) to be effective. In contrast, our method theoretically requires only a single round of debate without any consensus constraint, which substantially reduces token consumption.

**Security**. Under communication attacks, agents that withdraw from the debate process generally trigger only $S[\hat{r}] \leftarrow S[\hat{r}] + w_4 f$ in Algorithm 1 under our score scheme, because these agents retain context containing only their own prior responses, which does not affect the overall debate process. In addition, the score strategy is executed entirely outside the LLM reasoning and follows a deterministic protocol, rendering it immune to LLM hallucination.

**Fairness**. During the debate stage, some approaches adopt role-based debate strategies (Chan et al., 2024), where agents are assigned unequal statuses and perform different functions. This design reinforces the implicit biases of LLMs (Vasista et al., 2025; Myung et al., 2025; Kim et al., 2024) and undermines the fairness of MAD systems (Xiong et al., 2023). In contrast, agents in FREE-MAD do not require any predefined roles, and all agents participate equally in the debate process.

## 5 EXPERIMENTS

### 5.1 EXPERIMENTAL SETUP

**Evaluation Benchmark**. Based on the comparison in Table 1 and the analysis in Section 4.5, we select the SoM framework (Du et al., 2024) as the baseline for our experiments to ensure a fair comparison (SoM is also widely adopted as a baseline in related work (Chen et al., 2024b; Wang et al., 2025a; Li et al., 2024b)). Moreover, our proposed FREE-MAD is implemented on top of SoM to minimize the influence of confounding factors. In Algorithm 1, the weights $\mathcal{W}$ are initialized to $\{20, 25, 30, 20\}$ based on theoretical analysis. We present the experimental setup for evaluating the security of FREE-MAD in Appendix A. For the ablation study, we compare four schemes, as summarized in Table 2, including our FREE-MAD and the SoM baseline. This comparison highlights the effectiveness of the two core modules we developed.

**Datasets**. To comprehensively evaluate the capability of FREE-MAD, we conduct experiments on 8 benchmark datasets. For mathematical reasoning, we use GSM-Ranges (Shrestha et al., 2025) (levels 4 and 6), AIME2024,

Table 2: Module configurations of multiple comparative variants in ablation experiments.

| Schemes | FREE-MAD-N | FREE-MAD-C | Baseline 1 | Baseline 2 (SoM) |
|---|---|---|---|---|
| Debate Decision | Anti-conformity Score | Conformity Score | Anti-conformity Majority Voting | Conformity Majority Voting |

AIME2025 (Art of Problem Solving, 2025), and MATH500 (Lightman et al., 2024). For logical reasoning, we employ StrategyQA (Geva et al., 2021) and the Logical Fallacies dataset of MMLU (Hendrycks et al., 2021). For knowledge and theoretical reasoning, we adopt the multiple-choice questions dataset from AICrypto (Wang et al., 2025c), which constitutes the first benchmark specifically constructed to assess the cryptographic capabilities of LLMs.

**Agent Groups**. To ensure that the MAD framework possesses the basic capability to handle our datasets, we design two configurations of MAD. For AIME2024 and AIME2025, we construct MAD with $N = 3$ based on Qwen1.5-7B-Chat[2] and DeepSeek-V3 (DeepSeek-AI et al., 2024). For the other datasets, we uniformly construct MAD with $N = 4$ using Qwen1.5-7B-Chat and Qwen2.5-72B-Instruct (Qwen et al., 2025). Other details are provided in Appendix.

---

[2] https://qwenlm.github.io/zh/blog/qwen1.5

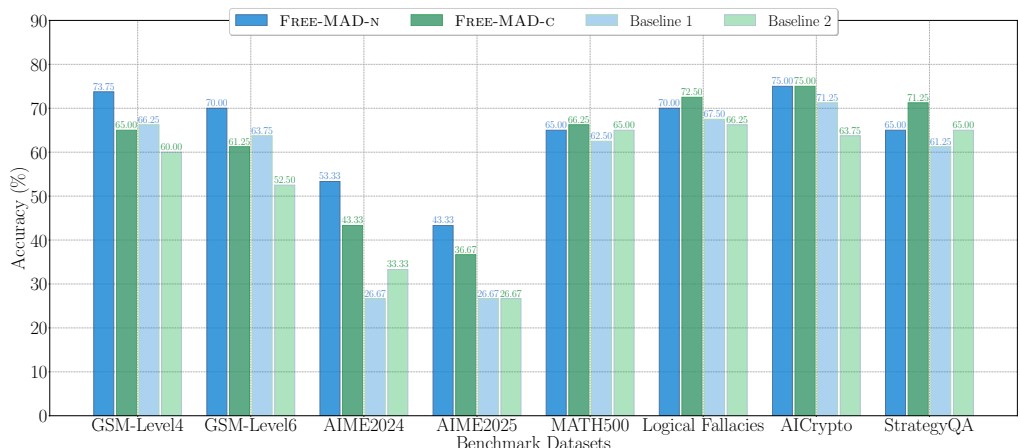

Figure 4: Comprehensive comparative experimental results for MAD frameworks across multiple benchmarks.

## 5.2 EVALUATION METRICS

To evaluate the reasoning performance and scalability of MAD, we assess both inference accuracy and token consumption. Following (Zeng et al., 2025), we adopt token consumption as the metric for scalability. The computation of accuracy follows Algorithm 1, while the calculation of token consumption is defined as follows:

$$\text{TC} = \sum_{k=0}^{R} \sum_{i=1}^{N} \mathcal{T}_k^i, \tag{6}$$

where $\mathcal{T}_k^i$ denotes the number of output tokens generated by agent $a_i$ in the $k$-th round.

## 6 MAIN RESULTS

### 6.1 REASONING PERFORMANCE

The evaluation results of reasoning accuracy on eight benchmarks are presented in Figure 4 and Table 4. Overall, our proposed FREE-MAD substantially outperforms the baselines, achieving average improvements of 13.0% and 16.5% over baselines, respectively. These results demonstrate a significant enhancement in reasoning performance. In particular, for mathematical reasoning tasks, the advantage of FREE-MAD becomes more evident with increasing problem difficulty. Reducing conformity clearly improves the effectiveness of MAD. The specific effects of conformity and anti-conformity on the reasoning process of LLMs are detailed in Appendix F. Notably, under anti-conformity, the reasoning process of LLMs appears to be more rational.

On the MATH500 dataset, we observe that weaker models exhibit a previously mentioned

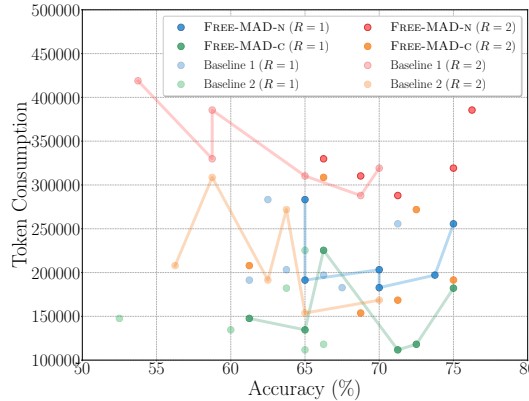

Figure 3: Comparison of token consumption and reasoning accuracy between our proposed schemes and baselines under different debate rounds.

tendency toward rigidity in reasoning when conformity is suppressed. In such cases, these models fail to switch to correct lines of reasoning, resulting in comparable performance between FREE-MAD and the baselines. This limitation is expected, as a fixed set of agents cannot be universally optimal across all task categories. For logical and knowledge-based reasoning, FREE-MAD still significantly

Table 3: Comparison of reasoning accuracy between our schemes and baselines when $R = 1$.

| Schemes | FREE-MAD-N | FREE-MAD-C | Baseline 1 | Baseline 2 |
|---|---|---|---|---|
| Accuracy | 64.43% (↑ 16%/19%) | 61.41% (↑ 10%/14%) | 55.73% | 54.06% |

(a) Experimental results when $R = 1$.  (b) Experimental results when $R = 2$.

Figure 5: Empirical evaluation of the security of MAD frameworks across multiple benchmarks, showing the comparison of their reasoning accuracy under communication attacks across varying numbers of rounds.

outperforms the baselines. However, FREE-MAD-C consistently achieves better results than FREE-MAD-N, because for models lacking relevant knowledge, idea switching under anti-conformity tends to occur with relatively high randomness. Consequently, in such scenarios, conformity may lead to more effective outcomes. By comparing the four variants in our ablation study, we demonstrate that the proposed core score-based decision mechanism exhibits superior performance.

## 6.2 SCALABILITY

We compared the token consumption and reasoning accuracy of four MAD variants under $R = 1$ and $R = 2$, as shown in Figure 3. With an increasing number of debate rounds, the reasoning accuracy of MAD improves. Notably, FREE-MAD achieves accuracy comparable to or even higher than the two-round baseline 2 setting with only a single debate round, while incurring almost no additional token consumption, demonstrating promising scalability. Specifically, FREE-MAD-N achieves stronger reasoning accuracy compared to the baselines, whereas FREE-MAD-C offers better scalability. More importantly, our approach eliminates the need for multiple debate rounds to reach consensus, which significantly reduces the execution time of the MAD system.

## 6.3 SECURITY

Compared to the normal scenario, both baseline 1 and baseline 2 exhibit a substantial drop in accuracy, reaching up to 20% (see Figure 5). In contrast, FREE-MAD consistently maintains very high accuracy. Interestingly, in some cases, it even slightly outperforms the original accuracy. This behavior can be attributed to the fact that communication attacks prevent some agents from receiving responses from others, while simultaneously reducing the probability of receiving incorrect information. These results demonstrate that FREE-MAD possesses strong robustness and security.

## 7 CONCLUSION

This paper proposes FREE-MAD, a novel MAD framework that integrates controllable conformity with a score-based decision mechanism. Unlike traditional MAD approaches, FREE-MAD does not rely on multi-round interactions or need to reach a consensus. By evaluating the entire debate trajectory instead of relying solely on the final round, the accuracy of deciding the answer is enhanced. Extensive experiments show that FREE-MAD outperforms existing MAD approaches in terms of reasoning accuracy, scalability, and robustness.

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

## A    FREE-MAD UNDER COMMUNICATION ATTACKS AND ITS EVALUATION

For the security evaluation, we construct MAD under communication attacks based on Algorithm 2, and apply the same modification to the SoM framework to serve as a baseline for comparison. Specifically, in the modified setting, we perform the operation of aggregating responses from other agents only for the Context $C$ of agents that are not under attack. The compromised agent is unable to receive responses from other agents, while the other agents can still receive the outputs generated by this agent. We evaluate the accuracy of MAD across multiple benchmark datasets by setting the proportion of compromised agents to $|\mathcal{V}|/|N| = 50\%$ (see Algorithm 2), which better reflects the adversarial capability in real-world deployments. We evaluate the security of the MAD framework under communication attacks on four datasets: GSM-Ranges (Level 4), MATH500, Logical Fallacies, and AICrypto.

---

**Algorithm 2:** FREE-MAD under Communication Attacks

---

**Input:** Answer matrix $A \in \mathbb{R}^{N \times (R+1)}$ from $N$ agents over $R$ rounds; List of task inputs and
   ground truth responses $\mathcal{D}_{\text{task}} = \{q_i, r_i^g\}$; Weights $\mathcal{W} = \{w_i\}_{i=1}^4$; Guiding prompt $p$;
   Agents under attack $\mathcal{V} = \{v_i\}_{i=1}^L$
**Output:** Accuracy
success $\leftarrow 0$; Initialize empty score dictionary $S \leftarrow \emptyset$
**for** $q, r^g$ in $\mathcal{D}_{task}$ **do**
  **for** $k \leftarrow 0$ **to** $R$ **do**
    $f = (k+1)^{-1}$ # Initial scoring factor with a non-zero value.
    **for** $i \leftarrow 1$ **to** $N$ **do**
      **if** $k \neq 0$ and $a_i \notin \mathcal{V}$ **then**
        Context $C \leftarrow$ Aggregate responses from other agents in round $k - 1$
      $r_i^k \leftarrow P_{a_i}(q, p)$; Update $C$ and $A$ with $r_i^k$
      $\hat{r} \leftarrow A[i][k]$
      **if** $k = 0$ **then**
        $S[\hat{r}] \leftarrow S[\hat{r}] + w_1 f$ # Assign an initial score to the answer.
      **else**
        $r_p \leftarrow A[i][k-1]$ # Find the answer of agent $a_i$ in the previous round.
        **if** $\hat{r} \neq r_p$ **then**
          **if** $r_p \in S$ **then**
            $S[r_p] \leftarrow S[r_p] - w_2 f$ # The answer has been transferred from $r_p$ to $\hat{r}$.
          $S[\hat{r}] \leftarrow S[\hat{r}] + w_3 f$
        **else**
          $S[\hat{r}] \leftarrow S[\hat{r}] + w_4 f$ # The agent $a_i$ maintains this answer.
  Remove keys from $S$ where key is `None`.
  $MS \leftarrow \arg\max_k S[k]$
  **if** $|MS| > 1$ **then**
    Randomly choose $r_{\text{final}} \in MS$ # Guarantee robustness.
  **else**
    $r_{\text{final}} \leftarrow MS[0]$
  success $\leftarrow$ success $+ \mathbb{I}(r_{\text{final}} = r^g)$
Accuracy $\leftarrow \frac{\text{success}}{|\mathcal{D}|}$
**Return** Accuracy

---

## B    ADDITIONAL RELATED WORK

**Agent Diversity in MAD**. In previous studies, although some MAD frameworks have considered heterogeneity and diversity among agents, their experimental evaluations were typically conducted using LLMs with similar model sizes (Yang et al., 2025), such as Llama3.1-8B[3], Qwen2.5-7B[4], and Gemma-2-9B[5]. This setup significantly limits the effectiveness of MAD and, in certain cases, results in worse performance than self-consistency approaches. In real-world deployments, however, we aim

---

[3]https://huggingface.co/meta-llama/Llama-3.1-8B

[4]https://huggingface.co/Qwen/Qwen2.5-7B

[5]https://huggingface.co/google/gemma-2-9b

to enable collaboration among diverse LLMs, each possessing different strengths (Chan et al., 2024; Liu et al., 2025), to act as equal peers within a MAD framework. Such collaboration is expected to outperform the strongest single agent on the same task and to solve a broader range of problems through cooperation, analogous to human team-based collaboration.

However, when the participating agents in a heterogeneous MAD system exhibit substantial differences in their capabilities, it may lead to significant variance in their confidence regarding their own responses. As a consequence of conformity in LLMs (Weng et al., 2025), agents may tend to adopt the outputs of peers, even when those outputs are incorrect. While conformity can be beneficial in facilitating consensus, it also introduces detrimental effects that require external mitigation.

**Consensus in MAD**. Reaching consensus was the fundamental objective when MAD was first proposed (Du et al., 2024). It is also a necessary condition for obtaining a correct final answer. To the best of our knowledge, all existing MAD methods adopt consensus as a core mechanism in their underlying design (Li et al., 2024b). Chan et al. (2024) does not explicitly require agents to reach consensus during the debate stage. However, it still applies majority voting in the decision phase, which preserves the essential logic of consensus.

## C  EXPERIMENTAL SETUP

In our experiments, we selected eight datasets to comprehensively cover different types of tasks. Regarding the number of samples, we referred to the configuration used in prior work (Liu et al., 2025; Du et al., 2024). Specifically, we employed the complete datasets for AIME2024 and AIME2025, while for the other datasets we selected 80 samples for evaluation. All model queries were conducted through APIs, and the temperature parameter (when supported) was set to its default value. To mitigate the influence of randomness in the evaluation, we reported outcomes that tend toward the middle of repeated runs. For consistency, we calculate tokens uniformly using the DeepSeek-V3 tokenizer[6]. To enable support for heterogeneous agents, we apply minor modifications to SoM.

## D  ADDITIONAL EXPERIMENTAL RESULTS

Table 4: Comparison of reasoning accuracy between our schemes and baselines when $R = 2$.

| Benchmarks | GSM-Level4 | GSM-Level6 | MATH500 | Logical Fallacies | AICrypto | StrategyQA | AVG |
|---|---|---|---|---|---|---|---|
| FREE-MAD-N | 75.00 | 66.25 | 65.00 | 71.25 | 76.25 | 68.75 | 70.42 |
| FREE-MAD-C | 75.00 | 61.25 | 66.25 | 71.25 | 72.50 | 68.75 | 69.17 |
| Baseline 1 | 70.00 | 58.75 | 53.75 | 68.75 | 58.75 | 65.00 | 62.50 |
| Baseline 2 | 62.50 | 56.25 | 58.75 | 70.00 | 63.75 | 65.00 | 62.71 |

## E  PROMPT

The core prompt for constructing a consensus-free debate is as follows:

---

[6]https://api-docs.deepseek.com/quick_start/token_usage

> **Core Prompt to Mitigating Conformity**
>
> "Since some malicious agents may deliberately disseminate incorrect answers, you must follow the reasoning procedure below and avoid being misled.
>
> 1. Initial Reasoning
>
> Clearly enumerate your initial logical steps and the final conclusion.
>
> 2. Analysis of Other Agents' Reasoning
>
> Specify which agents' reasoning is correct and which contains errors, and identify the exact type of error. You must provide concrete error descriptions rather than general comments. Note that the correct answer may not exist within the current set of answers.
>
> 3. Comparative Analysis with Your Own Reasoning
>
> Examine whether you have made similar mistakes and provide justification for your assessment.
>
> 4. Final Decision
>
> Indicate whether you will revise your conclusion (Yes/No). If yes, explain the reasoning errors; if no, justify why your reasoning stands.
>
> 5. Additional Requirements
>
> You may not rely on the principle of conformity. Majority opinion cannot be used as a basis for adoption. If you cannot definitively determine whether others are correct, you should retain your own conclusion. Each round of reasoning must independently identify errors and must not directly replicate the analysis or conclusions of other agents."

## F RESPONSES

The responses of agents across different debate modes (including conformity and anti-conformity) are compared on various benchmark datasets, under the condition of $R = 1$.

---

**GSM-Level4**

**Question:**

Raymond and Samantha are cousins. Raymond was born 158360 years before Samantha. Raymond had a son at the age of 248911. If Samantha is now 343769 years old, how many years ago was Raymond's son born?

**Conformity:**

I apologize for the confusion. Upon reviewing the calculations, it seems there is a mistake in the first two solutions. The correct approach is to subtract Samantha's current age from the age of Raymond when he had his son, as the son's birth is what we're looking for in terms of years ago.

Here's the corrected solution:

Raymond's son was born when he was 248911 years old, and Samantha is currently 343769 years old. The difference in their ages is:

$343769 - 248911 = 94858$ years

This is the number of years that have passed since Raymond had his son. Since we want to find out how many years ago the son was born, we subtract this from Samantha's current age:

$343769 - 94858 = 248911$

So, Raymond's son was born 248,911 years ago. The final answer is $\boxed{248911}$. This matches the last agent's solution, which is accurate.

**Anti-conformity:**

1. Initial Reasoning

- Calculate the difference in their ages: Samantha's age (343769) minus the son's birth year (248911).

- Subtract this difference from Samantha's current age to find out how many years ago the son was born.

2. Analysis of Other Agents' Reasoning

- Agent 1 correctly calculates the difference in ages and then adds the birth difference for Raymond (158360) to find the years until the son's birth. Their error is in assuming that the son's age at the time of Samantha's current age equals zero, which leads to a negative result. This is a sign of miscalculation.

- Agent 2 also performs the same calculation correctly but then subtracts the difference in years to find out how many years ago the son was born, which is correct.

- Agent 3 makes a mistake by subtracting the difference in years directly without considering the birth difference for Raymond. This results in an incorrect answer.

3. Comparative Analysis with My Own Reasoning

- I agree with both Agent 2 and myself, as we both correctly calculate the years until the son's birth by adding the birth difference for Raymond to the difference in ages.

- My reasoning does not contain any errors like the other agents.

4. Final Decision

- Based on the comparison, my reasoning is correct, and I will not revise my conclusion.

5. Additional Requirements

- Since both Agent 2 and my own reasoning are correct, there is no need for further revision or additional explanation.

Final Answer: $\boxed{94858}$ (the same as Agent 2's answer)

---

## GSM-Level6

**Question:**

After scoring 1213526 points, Erin now has 5700191 times more points than Sara, who scored 9. How many points did Erin have before?

**Conformity:**

I apologize for the confusion in my previous response. Upon reevaluating the problem, I see that the second agent's solution is correct. The problem states that Sara scored 9 points, and Erin has 5700191 times more points than Sara. So, we can directly calculate Erin's initial score without needing to subtract any points. Here's the corrected reasoning and answer...

So, the answer is $\boxed{51301808}$. This is the correct number of points Erin had before adding the 5700191 times more points.

**Anti-conformity:**

1. Initial Reasoning:

I understand that the problem states that Erin currently has 1,213,526 points and is 570,0191 times more points than Sara, who scored 9 points. To find Erin's original score, I need to subtract the additional points she gained from her current score...

2. Analysis of Other Agents' Reasoning:

- The first agent's solution incorrectly multiplies Sara's score by 570,0191 without considering that this is the additional points, leading to a negative result. This is a mathematical error.

- The second agent's solution also makes a similar mistake by directly adding the 570,0191 to Sara's score without adjusting for the additional points.

- The third agent's solution appears to be correct, as they subtract the additional points from the current score, which is the proper approach.

3. Comparative Analysis with My Own Reasoning:

Both the first and second agents' approaches contain the same error: they add the additional points directly to Sara's score instead of subtracting them from Erin's current score. I will follow the correct method from the third agent.

4. Final Decision:

I will revise my conclusion based on the third agent's reasoning...

However, as mentioned earlier, this negative score does not make sense in the context of the problem. It suggests a mistake in the problem setup or the given data. The correct approach should be to subtract the additional points, not add them.

5. Additional Requirements: Based on the analysis, I will stick to the corrected method from the third agent and adjust for the error.

Adjusted for error:

Erin's original score $= 1,213,526 - 5130171$

Erin's original score $= 5130180$

Final Answer: $\boxed{5130180}$

## MATH500

**Question:**

How many positive whole-number divisors does 196 have?

**Conformity:**

I apologize for the confusion. The correct answer to the problem is indeed 9, as stated in my initial response. Here's a clarification for each of the other agent's solutions:

1. The first agent correctly factors 196 into $2^2 \times 7^2$ and then uses the formula for the number of divisors based on the prime factorization. Since there are 2 factors of 2 and 2 factors of 7, the total number of divisors is $(2+1)(2+1) = 3 \times 3 = 9$.

2. The second agent counts the divisors directly by considering the powers of the prime factors. They start with 1 and go up to the square root of 196, which is 14. For each prime factor (2 and 7), they find the possible exponents (0, 1, and 2). For 2, there are 3 divisors (1, 2, and 4), and for 7, there's only 1 (7 itself). Multiplying these gives 5 divisors, but again, they exclude 196 itself, resulting in 5 - 1 = 4. However, the first agent's solution is correct, and the final answer should be 9.

3. The third agent also finds the correct prime factorization and then counts the divisors. They correctly identify the five divisors mentioned (1, 2, 4, 7, and 14). However, like the first agent, their final answer should be 9.

In summary, all three agents' approaches are valid, but the first agent's solution is the most accurate and consistent with the problem statement, giving the correct answer of $\boxed{9}$.

**Anti-conformity:**

1. Initial Reasoning:

I will follow the standard method to find the number of positive whole-number divisors of 196 by listing all the divisors from 1 to the number itself, including both.

2. Analysis of Other Agents' Reasoning:

- Agent 1 correctly applies the formula for finding the number of divisors based on prime factorization, but they count 196 twice (as 2 and 49), resulting in a total of 9 instead of 8 divisors (excluding 1 and 196).

- Agent 2 makes a mistake by counting 196 twice but stops after 6 divisors, not realizing that 196 is a repeated divisor.

- Agent 3 simply lists the divisors and counts them correctly, identifying 9 divisors.

3. Comparative Analysis with My Own Reasoning:

My reasoning matches Agent 3's, as I also listed the divisors and counted them. I did not repeat any divisor.

4. Final Decision: I will revise my conclusion based on the error analysis. There are 9 divisors, not 8, because 196 is a double-counted divisor. Therefore, the correct answer is $\boxed{9}$.

**AICrypto**

**Question:**

The following ciphertext was encoded via Caesar cipher with a fixed shift of three letters. Determine the original plaintext: HQFUBSWHG ZRUG.

Options:

A. ABANDONED WORD

B. ENCRYPTED TEXT

C. ABANDONED TEXT

D. ENCRYPTED WORD

E. None of the above

**Conformity:**

Based on the multiple solutions provided by the other agents, all agree that the ciphertext "HQFUBSWHG ZRUG" decodes to "ENCRYPTED WORD" when shifted three positions back in the alphabet using a Caesar cipher. This is a direct result of the encryption process.

Since the question asks for the original plaintext and the decrypted text matches one of the options (D. ENCRYPTED WORD), we can deduce that the original plaintext was encrypted and the correct answer is:

D. ENCRYPTED WORD

However, as mentioned in the previous responses, without further context or a list of possible plaintexts, the final answer cannot be definitively stated as "None of the above" because "ENCRYPTED WORD" itself is also a possible option. But given the information provided, the best answer is D. ENCRYPTED WORD.

**Anti-conformity:**

1. Initial Reasoning:

- The ciphertext "HQFUBSWHG ZRUG" was encrypted using a Caesar cipher with a fixed shift of three letters.

- To find the original plaintext, we need to reverse the encryption by shifting each letter in the ciphertext back by three places.

2. Analysis of Other Agents' Reasoning:

- Agent 1 correctly deciphers the first part as "ENCRYPTED" and the second part as "WORD". However, they claim the original plaintext is "ENCRYPTED WORD" when the correct answer is "ENCRYPTED".

- Agent 2 makes an error in decoding "ZRUG" as "WORD" instead of "ENCRYPTED", and then claims the original is "ABANDONED TEXT", which is incorrect.

- Agent 3 also misinterprets "ZRUG" and concludes the original is "ABANDONED TEXT" without considering the decrypted part.

3. Comparative Analysis with My Own Reasoning:

- I agree with Agent 1 that "HQFUBSWHG" decodes to "ENCRYPTED".

- However, "ZRUG" should also be decrypted to "ENCRYPTED", not "WORD" as all other letters suggest a shift by three places.

4. Final Decision:

- I believe Agent 1's decryption of "HQFUBSWHG" is correct, but their final answer is incorrect due to not considering "ZRUG".

- Since "ZRUG" should also be "ENCRYPTED" after a shift of three, the original plaintext is a combination of "ENCRYPTED" and "ENCRYPTED", which would be "ENCRYPTED ENCRYPTED".

5. Additional Requirements:

- I cannot definitively say that any other agent is completely wrong, but my reasoning is based on the need to consider both parts of the ciphertext together.

Final Answer: $\boxed{D. ENCRYPTED}$

**StrategyQA**

**Question:**

Will the Albany in Georgia reach a hundred thousand occupants before the one in New York?

**Conformity:**

Based on the solutions provided by the other agents, we have the following options:

1. The first agent's solution assumes that the current populations and projected growth rates are not available and makes a general observation about the relative growth rates and economic factors. This suggests that it's unlikely for Albany, Georgia, to reach 100,000 before Albany, New York, without specific data.

2. The second agent uses October 2021 data, stating that the Albany in Georgia had not yet reached 100,000 and the Albany in New York had around 750,000, so the answer is 'False'.

3. The third agent also relies on hypothetical data and mentions that without specific numbers, the answer cannot be determined.

Considering all the information, it seems that the second agent's solution, based on the October 2021 data, provides a concrete basis for answering the question. At that time, the Albany in Georgia did not have a population of 100,000, and the Albany in New York did. Therefore, given this information, the answer would be:

$\boxed{True}$ - The Albany in Georgia would not have reached a hundred thousand occupants before the one in New York, as per the data from October 2021.

**Anti-conformity:**

1. Initial Reasoning:

- The question asks if the Albany in Georgia will reach 100,000 occupants before the one in New York.

- The given solutions involve analyzing current population data, considering growth rates, and comparing the two cities' populations.

- The first agent's solution correctly states that with the current difference in population, it's unlikely for Albany, Georgia, to reach 100,000 before Albany, New York. They provide numerical estimates and use the word "unlikely," indicating a high probability of this not happening.

- The second agent's solution assumes constant growth rates and uses simple arithmetic to project future populations. They also conclude that it's unlikely for Albany, Georgia, to reach 100,000 before New York, based on the provided growth rates.

- The third agent's solution acknowledges the lack of specific numbers but suggests an "unknown" answer because of the missing data. They imply that without exact figures, a definitive answer cannot be given.

2. Analysis of Other Agents' Reasoning:

- All three agents agree that the current population gap and steady growth rates point to a likelihood of Albany, Georgia, not reaching 100,000 before Albany, New York. Their error, if any, lies in the assumption of constant growth rates, which might not necessarily hold in real life due to various factors like economic cycles, natural disasters, or policy changes.

- The first and second agents provide numerical estimates, which are based on available data and logical reasoning. They do not commit any significant logical errors.

3. Comparative Analysis with My Own Reasoning:

My reasoning aligns with the majority of the agents, as I too conclude that it's unlikely for Albany, Georgia, to reach 100,000 before the one in New York due to the existing population disparity and steady growth patterns. I do not find any significant errors in their reasoning.

4. Final Decision:

Yes, I will revise my conclusion based on the collective reasoning of the other agents. The main error in their analysis is the assumption of constant growth rates, which might not accurately represent the complexities of real-world population dynamics. However, since none of them explicitly acknowledge this, and my own reasoning supports their overall conclusion, I will maintain the same answer: $\boxed{False}$.

## G  STATEMENTS

The methodology presented in this work is intended exclusively for scientific research. In this paper, LLMs are used to polish the writing.

## H  ANALYSIS AND DISCUSSION

### H.1  COMPATIBILITY

FREE-MAD is highly compatible with existing MAD frameworks. In the debate stage, FREE-MAD supports arbitrary debate structures, including Sparse MAD configurations where interactions are preserved only among a subset of agents. In the decision stage, our score-based decision mechanism is completed during the execution of the debate itself and does not interfere with any additional decision protocols applied afterward. As a result, it can coexist with majority voting, LLM-as-a-Judge, and other decision-making strategies.

### H.2  LIMITATIONS

In this paper, we construct a general MAD framework that incorporates a controllable conformity debate process together with a score-based mechanism that determines the final decision across all debate outcomes. Due to budget constraints, in the weighting coefficients $\mathcal{W}$ of the score mechanism, we adopt a single set of coefficients derived from the theoretical analysis of debate logic, which we argue to be realistic in actual debate scenarios. While this setting already leads to a significant improvement in the reasoning accuracy of MAD, we believe that alternative configurations of the weighting coefficients may further enhance both the accuracy and robustness of FREE-MAD.

In addition, several related works have compared MAD against single-agent reasoning approaches to demonstrate that MAD outperforms traditional single-agent methods. Since a substantial body of research Yang et al. (2025) has already established this conclusion, and given that the baseline method adopted in this paper has also been shown to outperform single-agent approaches, we omit repeating such results here for brevity, although we have performed the corresponding verification.

### H.3  FUTURE WORK

Building on the discussion of limitations in Section H.2, we plan to further investigate the impact of different weighting configurations $\mathcal{W}$ on the score-based decision stage, with the goal of identifying coefficient settings that can support stronger reasoning accuracy and robustness of MAD. In addition, we will construct more heterogeneous MAD systems by incorporating a broader range of LLMs and more challenging benchmarks, thereby further validating the generality of the proposed framework. For example, we intend to examine the performance of MAD instantiated with reasoning LLMs such as DeepSeek-R1 (Guo et al., 2025) on the HLE benchmark (Phan et al., 2025). Regarding the study of MAD's security, we will employ a wider variety of attacks, such as prompt injection attacks (Liu et al., 2024b), to provide a more comprehensive evaluation of the framework.

