# OpenReview forum: "Free-MAD: Consensus-Free Multi-Agent Debate"
_ICLR.cc/2026/Conference — ICLR 2026 Conference Withdrawn Submission_

### Official Review · Reviewer_h57Z · 2025-10-28

**Soundness:** 2
**Presentation:** 3
**Contribution:** 3
**Rating:** 4
**Confidence:** 4

**Summary:**

The paper proposes FREE-MAD, a consensus-free multi-agent debate framework for LLM reasoning.  Instead of the traditional rounds of debate, followed by a majority vote at the end, as presented in MAD; the authors present present a method that: (1) redesigns the debate stage with an anti-conformity, CoT-driven prompt that asks agents to only change their answer when they can justify that their previous answer was wrong; and (2) replaces final-round voting with a score-based decision mechanism that aggregates all intermediate answers across rounds. Experiments on 8 benchmarks shower higher accuracy results than the baselines, with added rousted and reduced token cost.

**Strengths:**

1. The paper a new MAD protocol decomposed in 2 phases. The method moves away from consensus and integrates anti-conformity.
2. The method shows achieving competitive performance for single-round debate, and better performance in some cases for multiple rounds.
3. The paper clearly presents the method, including accurate descriptions and the algorithm's underlying principles and implementation details

**Weaknesses:**

1.The authors assume that answer switch between rounds leads to an improvement. However, it doesn't take into account the change could lead to worse performance. The algorithm could take into account the quality improvement of those answer switches: fraction of switches that move toward ground truth per round
2. The robustness analysis focuses on communication attacks, but it misses to analyze some more realistic attacks, such as contamination, adversarial attack or colluding agents. The robustness claims seem to be overstated given the lack of analysis.
3. Similarly, the authors make claim regarding scalability of their method, but only perform experiments for up to 2 rounds, how does this method behave with larger number of rounds.

**Questions:**

- The baselines are not clear. The paper should include a paragraph describing the baselines. To make it clearer, it is better to refer to them by their name rather than by writing Baseline 1 or Baseline 2.
- Why doesn't Figure 4 clearly state the number of rounds for that experiment?
- Why is Figure 3 presenting the results Token Consumption vs Accuracy? If the purpose of this plot is to show scalability as mentioned in Section 6.2, wouldn't it be better to present Token Consumption and Accuracy vs Number of Rounds. I think the current plot is not presenting new information.
- Concerns about robustness. The authors have selected the communication attack, but have neglected some more real attacks such as contamination, adversarial attack or colluding agents. Can they prove their method is robust against other type of attacks?
- How much does the method scales? How many agents or rounds can the method scale to and what are the benefits?

---

> ### Author Response · Authors · 2025-11-17
>
> We sincerely thank reviewer h57Z for acknowledging the performance of our method as well as the clarity of our algorithmic principles and implementation details. We address each concern as follows.
>
> > It doesn't take into account the change could lead to worse performance.
>
> 1. We appreciate the reviewer for pointing out that answer changes between rounds may potentially harm performance. We would like to clarify that, as shown in Algorithm 1, our reward formulation not only encourages answer switches but also preserves the previous-round answer when appropriate, achieving a balance between exploration and stability. Moreover, because models become increasingly vulnerable to conformity as the number of rounds R grows, we introduce a correction factor that reduces the weight of later-round rewards in order to mitigate the impact of erroneous switches.
>
> > The algorithm could take into account the quality improvement of those answer switches.
>
> 2. Regarding the reviewer’s suggestion to measure the fraction of switches that move toward the ground truth per round, we note that in MAD settings the agents do not have access to the ground truth during debate. Ground-truth labels can only be used after the decision stage to compute accuracy, not during the iterative reasoning process.
>
> > The baselines are not clear.
>
> 3. We would like to emphasize that the baselines and their full experimental configurations are described in detail in Section 5.1 (lines 353–360) and in Table 2. We will further improve the clarity of baseline naming in the revised version.
>
> > Why doesn't Figure 4 clearly state the number of rounds for that experiment?
>
> 4. Figure 4 reports results under R=1. The corresponding results for R=2 are included in Table 4. We will make this explicit in the revised manuscript.
>
> > Why is Figure 3 presenting the results Token Consumption vs Accuracy?
>
> 5. We appreciate the reviewer’s question regarding Figure 3. In fact, we have already provided a detailed analysis in Section 6.2 (lines 461–468). Conceptually, Figure 3 presents exactly what the reviewer suggests (Token Consumption and Accuracy versus the Number of Rounds) and includes results for both R=1 and R=2. Compared to other experimental results, Figure 3 specifically highlights the relationship between token consumption and accuracy, thereby illustrating the scalability characteristics of our method.
>
> > Concerns about robustness.
>
> 6. We appreciate the reviewer’s suggestions for additional attack types. However, we would like to clarify several points:
> (1) In Section 3.2 (lines 187–192), we provide a principled analysis of MAD robustness with a focus on multi-agent interaction security, which is directly relevant to MAD systems. The attacks mentioned by the reviewer, such as contamination, adversarial examples, or collusion, are traditionally designed for single-agent or distributed systems, not MAD frameworks.
> (2) In realistic MAD deployments, agents are typically randomly sampled from a model pool, significantly reducing the likelihood of these coordinated attacks.
> (3) As shown in Table 1, FREE-MAD does not rely on LLM-as-a-Judge, which inherently lowers susceptibility to contamination or adversarial judge manipulation.
> (4) Moreover, the communication attack used in our experiments is, to the best of our knowledge, the only peer-reviewed attack that is applicable to MAD in real-world deployment settings.
>
> > How many agents or rounds can the method scale to and what are the benefits?
>
> 7. From the perspective of MAD research, the central goal is to achieve higher accuracy using fewer models and fewer debate rounds, not to scale MAD to arbitrarily large groups of agents. FREE-MAD is designed around this principle, achieving strong performance even under a single round (R=1), outperforming other methods that require multiple rounds. Moreover, due to the inherent context length limitations of the models and practical token consumption considerations, our analysis in Section 4.5 (lines 321–330) shows that token usage grows quadratically with N, making MAD with an excessively large number of agents impractical.
>
> We again thank reviewer h57Z for the constructive suggestions, which have helped us strengthen the paper. We are happy to address any further questions.

---

> > ### Comment · Reviewer_h57Z · 2025-11-27
> >
> > Regarding taking into account quality improvement of answer switches, I referred to using the score computing in the middle rounds.
> >
> > With respect to the baselines, I think it would be easier to refer to them by their names in the plots and tables, instead of Baseline 1 or Baseline 2.
> >
> > Why do you not include the results for different number of rounds in the main section of the pape? For instance, in one of the main papers in the literature of MAD, they evaluate over 3 agents and 3 rounds of debate. How is your method affected and compared with the baselines when the number of rounds is increased to at least R=3? I think this should be included in the main section of the paper.
> >
> > In the current form, it is still unclear what is driving the performance improvement and how reliable this improvement is when compared to other SoTA methods in the MAD under fair comparison conditions.
> >
> >
> > [1] Du, Yilun, et al. "Improving factuality and reasoning in language models through multiagent debate." Forty-first International Conference on Machine Learning. 2023.

---

> > > ### Author Response · Authors · 2025-11-27
> > >
> > > We thank Reviewer h57Z for the response, and we further address the raised concerns below.
> > >
> > > 1. We appreciate the reviewer’s additional clarification regarding taking into account the quality improvement of answer switches. In fact, as shown in Algorithm 1, score computing in the middle rounds has already been incorporated. However, the goal of our work is to accomplish an accurate debate with the minimum number of rounds, thereby reducing the substantial token cost commonly observed in existing approaches. Moreover, since we have already achieved the optimal setting of R = 1, the middle round is effectively the final round.
> > >
> > > 2. We will revise the description of the baselines according to the reviewer’s request.
> > >
> > > 3. We thank the reviewer for the question. Previously, we placed these results in the appendix because our work aims to address a fundamental issue in existing MAD frameworks, the need for multiple debate rounds to reach consensus, which leads to extremely high token consumption. Our focus is therefore on achieving high accuracy with as few rounds as possible, which is why we primarily reported results for small value of R. In the revised version, we will include the results for different numbers of rounds in the main section of the paper.
> > >
> > > In addition, we thank the reviewer for providing reference [1]. In fact, this reference is one of the baselines used in our work. According to this reference, R = 3 is the upper limit for debate rounds in MAD. The definition of R in that paper differs from ours: their R = 3 corresponds to R = 2 in our formulation. These classical MAD approaches typically require multiple debate rounds to reach consensus to ensure accuracy. In contrast, our method does not target consensus; it can achieve high accuracy with just a single round of debate.
> > >
> > >
> > > 4. Regarding the reason for driving the performance improvement, our ablation study has already provided the requested comparisons. Table 2 and Figure 4 compare four variants: two baselines using majority voting and two FREE-MAD variants using the same debate prompts but different decision rules. These variants allow isolation of (1) debate prompt effects and (2) our decision mechanism. Comparing FREE-MAD-N with Baseline 1 isolates the impact of the score mechanism under identical anti-conformity prompts; comparing FREE-MAD-C with Baseline 2 (SoM) isolates anti-conformity when the decision mechanism is replaced with majority voting. Across all datasets, the improvement persists even when prompts are held constant, demonstrating that the gains cannot be attributed merely to shifted prompting or increased awareness of reasoning gaps. Regarding the possibility that improvements come only from updated prompts, we reiterate that Baseline 1 and Baseline 2 already control for this: Baseline 1 uses exactly the same anti-conformity prompts as FREE-MAD-N, yet performs substantially worse; Baseline 2 uses the same conformity prompts as FREE-MAD-C, yet again performs worse. This directly confirms that the scoring mechanism is the primary contributor. Moreover, to the best of our knowledge, since the publication of reference [1], MAD approaches have universally adhered to consensus as the underlying mechanism. Subsequent improvements have primarily focused on optimizing the upper-level agents, such as reducing token consumption by shifting from fully connected to sparsely connected structures. These later studies are orthogonal to our work. FREE-MAD targets optimizations at the underlying consensus mechanism and can likewise support and accommodate upper-level improvements. Regarding compatibility, we have already added new code in the supplementary materials.

---

> ### Author Response · Authors · 2025-11-27
>
> We sincerely thank Reviewer h57Z for the constructive comments that helped improve our work. We would be glad to address any further questions.

---

### Official Review · Reviewer_vKaB · 2025-10-30

**Soundness:** 2
**Presentation:** 3
**Contribution:** 2
**Rating:** 4
**Confidence:** 3

**Summary:**

The paper presents FREE-MAD, a framework for multi-agent debate that improves how LLMs reason together without needing consensus. Instead of using majority voting, it scores all agents’ responses across the debate to find the most consistent and well-reasoned answer. By encouraging anti-conformity, agents critique rather than copy one another, reducing groupthink and errors. Experiments on eight benchmarks show that FREE-MAD improves accuracy, scalability, and robustness while needing only a single debate round

**Strengths:**

1. By leveraging anti-conformity, FREE-MAD encourages agents to question and critique others’ reasoning rather than follow the majority, reducing groupthink and improving the overall accuracy

2. It achieved notable performance gains while reducing token consumption through just a single round of debate.

3. Good writing and easy to understand.

**Weaknesses:**

1. The score-based decision mechanism employs a fixed set of weighting coefficients derived from theoretical analysis. However, it seems to lack a clear analytical rationale or process explaining how these values were determined.
2. In the agent groups, when N = 3 or N = 4, the setup is based on only two models — Qwen1.5-7B-Chat and DeepSeek-V3, or Qwen1.5-7B-Chat and Qwen2.5-72B-Instruct. This design is somewhat confusing, as it’s unclear which model each agent corresponds to. In addition, the variety of agent types is limited, since the configuration relies on only two model architectures.
3. The paper doesn’t include a clear analysis showing how key parameters—like the scoring weights or number of agents—affect performance or stability.

**Questions:**

1. Is there a clear analytical or empirical basis for choosing the specific coefficient values used in the model?
2. Additional experiments should be conducted using a more diverse set of models and a larger range of N values.

---

> ### Author Response · Authors · 2025-11-17
>
> We sincerely thank reviewer vKaB for the recognition of our method’s accuracy, efficiency, and clarity of presentation. Below we address each concern in detail.
>
> > It seems to lack a clear analytical rationale or process explaining how these values were determined.
>
> 1. Regarding the weighting coefficients, we appreciate the reviewer for raising this point. As discussed in Section H.2 (lines 1204–1216), the coefficients used in our method were selected based on extensive empirical observations as well as theoretical considerations of the scoring design. Furthermore, in Section H.3 (lines 1219–1227), we explicitly acknowledge that investigating the impact of alternative weighting configurations W on the score-based decision stage is a future research direction. Despite this, our experiments across a wide range of task datasets demonstrate that the current coefficient configuration shows strong generality and consistently outperforms all baselines. This validates its effectiveness for the scope of this work.
>
> > This design is somewhat confusing.
>
> 2. We would like to clarify that influential prior MAD studies commonly adopt N = 3 or N = 4 as the standard setup. Our choices of N, model types, and model sizes follow these established conventions for fair comparison. We agree that the correspondence between agents and model instances should be described more explicitly, and we will provide a clearer explanation in the revised version to avoid confusion.
>
> > The variety of agent types is limited. Additional experiments should be conducted using a more diverse set of models and a larger range of N values.
>
> 3. From the long-term perspective of MAD research, the key objective is to achieve higher accuracy with fewer models and fewer debate rounds, rather than scaling to a large number of agents. Due to inherent context length limitations and practical token cost concerns, increasing N becomes infeasible. As analyzed in Section 4.5 (lines 321–330), token consumption grows quadratically with N, making MAD with excessively large agent groups impractical. When N=5, in practice, for typical models, the debate history already exceeds the model’s context length limit when R=2. This results in truncation and consequently leads to reasoning errors.
> We also agree with the reviewer that agent diversity can further enhance MAD performance. This observation is aligned with our analysis in the Appendix (lines 803–813). However, we emphasize that if the available model pool already contained sufficiently diverse models, then MAD would not be necessary in the first place, which contradicts the fundamental motivation.
>
> We sincerely thank reviewer vKaB for the constructive comments that helped improve our work. We would be glad to address any further questions.

---

### Official Review · Reviewer_keYr · 2025-10-31

**Soundness:** 2
**Presentation:** 3
**Contribution:** 2
**Rating:** 2
**Confidence:** 3

**Summary:**

This paper argues that standard multi-agent debate (MAD) suffers from conformity, majority-vote randomness, and multi-round token overhead, and proposes FREE-MAD—a consensus-free framework that evaluates the entire debate trajectory instead of only the last round. The method reconstructs both stages of MAD: (i) a debate stage that can encourage anti-conformity via a CoT-style prompt to reduce error propagation, and (ii) a score-based decision stage that maintains a matrix of agents’ answers across rounds and updates a score dictionary to reward newly adopted answers and downweight abandoned ones, with a round-dependent factor to limit late-round conformity. The authors formalize the MAD protocol, compare against majority voting and LLM-as-judge designs, and introduce two practical variants—FREE-MAD-N (anti-conformity debate) and FREE-MAD-C (conformity debate)—that share the same scoring mechanism. Across eight benchmarks, FREE-MAD delivers sizable gains, with average improvements of roughly 13–16.5% over baselines, and remains competitive or better even when baselines use more rounds.

**Strengths:**

* Conformity is a critical challenge in the development of MAD, which motivates FREE-MAD well.
* FREE-MAD achieved significant improvement on diverse benchmarks

**Weaknesses:**

* There are too many hyperparamters in FREE-MAD, including w1-w4 and f to manage the behavior of scoring. The selection of hyperparameters are not analyzed.
* The scoring mechanism proposed in this paper is not "consensus-free". It is more like an advanced consensus mechanism considering longer context. The name is misleading and overclaiming.
* In Section 6, "Consequently, in such scenarios, conformity may lead to more effective outcomes." However, FREE-MAD is motivated by mitigating the conformity during MAD to improve the outcome. This observation seems contradictary to the motivation, and is not well explained.
* From the analysis, I cannot be convinced that the new "consensus-free debate stage" is the root cause for the improvement. There is a significant lack of detailed anaylsis to validate the effectiveness of consensus-free debate stage, including find-grained ablation study and comparison to baselines. For instance, is it possible that the improvement is simply brought by shifted focus during the debate due to the updated prompts, which makes LLMs aware of reasoning gaps?

**Questions:**

See weaknesses.

---

> ### Author Response · Authors · 2025-11-17
>
> We sincerely thank the reviewer keYr for the thoughtful feedback. Below we address each concern in detail.
>
> > The selection of hyperparameters are not analyzed.
>
> 1. Regarding the hyperparameters, these values are derived from the theoretical design of the scoring mechanism and extensive experimental validation, and are intended to be fixed, interpretable components rather than tunable variables. FREE-MAD follows the training-free paradigm of existing MAD frameworks. The same configuration is used across all eight benchmarks, yet yields consistent gains, reflecting the generality of the existing hyperparameters. We will clarify this philosophy in the revised version.
>
> > The name is misleading and overclaiming.
>
> 2. Concerning the naming of “consensus-free,” we respectfully clarify that our method does not implement an advanced form of consensus, nor does it rely on agreement among agents at any stage. FREE-MAD eliminates the requirement for consensus in both debate and decision: it does not use majority voting, does not encourage convergence during debate, and remains effective even when agents’ final-round responses remain divergent. The scoring mechanism evaluates the trajectory of reasoning rather than similarity across agents.
>
> > This observation seems contradictary to the motivation, and is not well explained.
>
> 3. The reviewer points out a perceived contradiction between our motivation of mitigating harmful conformity and our observation in Section 6 that conformity may sometimes help. Our motivation is to mitigate uncontrolled conformity, which often harms MAD by causing incorrect answers to propagate. In specific knowledge-intensive tasks, however, very weak agents may become overly stochastic when conformity is strictly suppressed, and limited conformity can stabilize their reasoning. This observation is precisely why our framework includes both FREE-MAD-N and FREE-MAD-C, enabling task-adaptive control. We have discussed this in detail in Section 4.4 (lines 298–318).
>
> > There is a significant lack of detailed anaylsis to validate the effectiveness of consensus-free debate stage.
>
> 4. The reviewer states that there is a lack of ablation to validate the effectiveness of the consensus-free debate stage. However, our ablation study has already provided the requested comparisons. Table 2 and Figure 4 compare four variants: two baselines using majority voting and two FREE-MAD variants using the same debate prompts but different decision rules. These variants allow isolation of (1) debate prompt effects and (2) our decision mechanism. Comparing FREE-MAD-N with Baseline 1 isolates the impact of the score mechanism under identical anti-conformity prompts; comparing FREE-MAD-C with Baseline 2 isolates anti-conformity when the decision mechanism is replaced with majority voting. Across all datasets, the improvement persists even when prompts are held constant, demonstrating that the gains cannot be attributed merely to shifted prompting or increased awareness of reasoning gaps. Regarding the possibility that improvements come only from updated prompts, we reiterate that Baseline 1 and Baseline 2 already control for this: Baseline 1 uses exactly the same anti-conformity prompts as FREE-MAD-N, yet performs substantially worse; Baseline 2 uses the same conformity prompts as FREE-MAD-C, yet again performs worse. This directly confirms that the scoring mechanism is the primary contributor.
>
> We appreciate the reviewer’s time and constructive comments. We hope these clarifications resolve the concerns and are happy to address any further questions.

---

### Official Review · Reviewer_z4VN · 2025-11-01

**Soundness:** 2
**Presentation:** 3
**Contribution:** 2
**Rating:** 4
**Confidence:** 4

**Summary:**

The paper proposes FREE‑MAD, which replaces final‑round majority voting with trajectory‑level scoring. During debate, prompts ask agents to revise only when they can falsify their prior answer. During decision, the method treats whether an agent revised or persisted in each round as signals for weighted scoring, with a late‑round decay, and returns the highest‑scoring answer. Experiments on math/logic/QA benchmarks show gains over majority‑vote baselines.

**Strengths:**

The method is simple: it uses revision vs. persistence as signals to realize a multi‑round weighted voting scheme.

Empirical results show improvements.

**Weaknesses:**

Code is not open-sourced, hindering reproducibility.

The motivation (that consensus/conformity may dilute or derail correct reasoning) is supported mainly by illustrations and anecdotal observations, not a direct statistical analysis.

The core method rewards revisions (up‑weighting the new answer and down‑weighting the old answer) and gives a smaller reward to persistence. In essence, this encourages changing answers. Although the authors claim the algorithm promotes evidence‑based rather than conformity‑based revisions, it does not actually detect the cause of a change; meanwhile the motivation discourages conformity—this creates a conceptual inconsistency.

The approach applies to tasks with a single, explicit correct answer (e.g., QA, MATH). It does not directly transfer to more complex coding or agentic tasks.

**Questions:**

For Limitation 1, is there a transfer path to coding/agentic tasks?

Will you compare against Self‑Consistency [1]? There are recent studies showing that Self‑Consistency typically outperforms MAD in both cost and accuracy [2, 3].

[1]  Self‑consistency improves chain‑of‑thought reasoning in language models.

[2] If multi‑agent debate is the answer, what is the question.

[3] Should we be going MAD? A Look at Multi-Agent Debate Strategies for LLMs

---

> ### Author Response · Authors · 2025-11-17
>
> We sincerely thank Reviewer z4VN for the insightful and constructive feedback. We address the concerns below.
>
> > Closed-source limits reproducibility.
>
> 1. We would like to clarify that all code necessary for reproduction has already been provided in the Supplementary Material. To comply with the anonymity requirements during the review period, we did not include links to an external public repository in the submission. We will add these links in the revised version to further facilitate reproducibility.
>
> > Motivation rests on anecdotes, not statistics.
>
> 2. Regarding the concern that our motivation relies mainly on illustrations or anecdotal observations: in fact, we provide a detailed analysis in the Introduction (lines 47–53). Moreover, this motivation is strongly supported by prior literature, including Wang et al., 2025b; Weng et al., 2025; Zhu et al., 2025; and Cho et al., 2025. The illustrations were included only to give readers more intuitive examples of these well-documented issues.
>
> > The core method rewards revisions…conceptual inconsistency.
>
> 3. We thank the reviewer for this insightful observation. We would like to clarify that, as shown in Algorithm 1, our scoring scheme encourages both revising and persisting with previous answers, striking a balance rather than simply promoting changes. As the reviewer noted, later rounds receive a decayed weight, which helps reduce conformity-driven switching. In addition, identifying why an agent revises its answer is not the focus of existing MAD methods nor of our work. The key signal used across these approaches is the act of revising itself. It is precisely under anti-conformity that, based on our design of the debate process, an agent’s change of answer can better reflect evidence-based reasoning.
>
> > Unclear transfer to coding/agentic tasks.
>
> 4. We have already noted in the Introduction (lines 44–45) that MAD can be applied to various scenarios, such as software issue resolution (Li et al., 2025), mathematical reasoning (Zhang & Xiong, 2025), and code summarization (Chun et al., 2025). This shows that MAD can be easily extended to other applications. At the same time, we agree with the reviewer that the method does not directly transfer to more complex tasks. This is because, to the best of our knowledge, all existing MAD approaches (e.g., Du et al., 2024; Zeng et al., 2025; Liang et al., 2024) follow the assumption (only one option can be selected), which is also conducive to multi-round, large-scale experiments. For a fair comparison, we also adhere to this assumption, and we have already clarified this in the footnote on page 3.
>
> > Recent work shows Self-Consistency often outperforms MAD.
>
> 5. We appreciate the reviewer raising this question and providing relevant references. As discussed in detail in the Appendix (lines 1213–1216), our chosen baselines have been shown in multiple prior works to outperform Self-Consistency. Papers [2] and [3] observe that MAD may underperform Self-Consistency in some settings, but as [2] explains, this is primarily because the compared MAD variants rely on homogeneous agents, which restricts MAD’s advantages. With heterogeneous agents, MAD generally outperforms Self-Consistency. For example, in [2], heterogeneous MAD achieves higher accuracy on MMLU and AGIEval (83.5, 70.1) compared to Self-Consistency (82.13, 67.07). We further examine the importance of agent diversity in Appendix (lines 803–813). Since our method is built on heterogeneous MAD and is recognized by the reviewer as clearly outperforming the baselines, our experimental results provide strong evidence of its effectiveness.
>
> We sincerely thank Reviewer z4VN again for helping us improve the quality of the paper. We would be grateful for any further comments or questions.

---

> > ### Comment · Reviewer_z4VN · 2025-11-26
> >
> > Thank you for the clarification; I have now reviewed your released code. As you yourself have noted, since your method cannot generalize to a broader range of practically valuable tasks, I am inclined to maintain my current score.

---

> > > ### Author Response · Authors · 2025-11-26
> > >
> > > We thank Reviewer z4VN for the feedback. We would like to clarify that the generalizability of Free-MAD as a theoretical method to a broader range of practical tasks largely depends on the specific technical implementation. Our previously released code followed the conventional MAD framework and thus had certain limitations in application scenarios.
> > >
> > > Recently, however, independent researchers from the research community have reproduced Free-MAD based on our paper and further extended it into an industrial-grade, fully deployable platform. Their implementation has been fully open-sourced. Notably, they made all system hyperparameters, including weights, adaptively configurable and introduced several normalization-based engineering strategies, which substantially enhance the system’s practicality and parallelization capacity. The resulting system is highly adaptive and can be applied to arbitrary scenarios.
> > >
> > > To preserve anonymity, we have anonymized their codebase and included it in the supplementary materials for reviewers’ examination. The supplementary files also provide a demonstration video showcasing industrial-level applications across five advanced LLMs: gpt-5-codex-mini, gpt-5-codex, gpt-5-high, claude-opus, and claude-sonnet. We will incorporate these valuable contributions from the research community into the revised version of the paper, presenting their platform as the industrial-grade implementation of the Free-MAD codebase, alongside our own original experimental version.
> > >
> > > We again thank Reviewer z4VN for the careful review, and we would be happy to address any further questions.

---

### Author Response · Authors · 2025-11-26
**Industrial-grade Implementation of Free-MAD**

We are pleased to report that independent researchers from the research community have recently reproduced Free-MAD based on our paper and further expanded it into an industrial-grade, fully deployable platform. Their implementation has been fully open-sourced. Notably, they make all system hyperparameters, including weights, adaptively configurable, and introduce normalization-based and several additional engineering strategies that substantially improve the system’s practicality and parallelization capacity. The resulting system is highly adaptive and can be applied to arbitrary scenarios, which directly alleviates the reviewers’ concerns regarding hyperparameter selection and applicability across diverse settings.

To preserve anonymity, we have anonymized their codebase and included it in the supplementary materials for reviewers’ examination. The supplementary files further provide a demonstration video showcasing industrial-level applications across five advanced LLMs: gpt-5-codex-mini, gpt-5-codex, gpt-5-high, claude-opus, and claude-sonnet. We will also incorporate this valuable contribution from the research community into the revised version of the paper. Specifically, we present their platform as the industrial-grade implementation of the Free-MAD codebase, alongside our own original experimental version.

---

### Author Response · Authors · 2025-11-28

Dear Reviewers and Area Chair,

We are well aware that the recent issues with the OpenReview system may have caused concern for the reviewers. We fully understand and deeply appreciate the time and effort you have devoted to our submission. We’re all navigating this frantic world side by side. Although we no longer hold any expectations regarding potential score adjustments, we sincerely hope that the reviewers can continue to provide us with further suggestions to help improve the quality of our paper. We aim to meet the attention and needs of the research community and to develop FREE-MAD into a general, industry-grade platform that can benefit a wide range of researchers, rather than limiting it to a single publication.

We would like to express our heartfelt gratitude to all reviewers and Area Chair for your dedication.

---

### Note · Authors · 2026-01-02

I have read and agree with the venue's withdrawal policy on behalf of myself and my co-authors.